# Improving Dual-Output Learning through Sparse Autoencoder Guided Disentanglement in Speech Representations

## Abstract

While representation disentanglement is often studied from the perspective of interpretability, its role in managing competing objectives in end-to-end learning remains less explored. We study dual-output automatic speech recognition (ASR), where a shared Conformer encoder supports both surface-level and meaning-oriented transcriptions, leading to representational interference. To address this, we propose SODA, an alignment-aware dual-output ASR framework that integrates a sparse autoencoder (SAE) as a trainable intermediate component. By applying auxiliary connectionist temporal classification (CTC) supervision at both the encoder output and the SAE latent space, the framework guides the formation of sparse, task-relevant representations consumed by two Transformer-based decoders. Experiments demonstrate that SODA performs competitively on both tasks within a single model. Our framework requires significantly fewer parameters than comparable baselines and learns interpretable, task-specific representations.

## 1. Introduction

Recent advances in automatic speech recognition (ASR) have led to impressive transcription accuracy. These advances are driven by modern architectures such as attention-based encoder–decoders and Conformer models (Gulati et al., 2020; Baevski et al., 2020; Watanabe & et al., 2021). By leveraging contextual information, ASR models can normalize pronunciation variability and produce linguistically plausible outputs even under noisy or imperfect conditions.

Despite this progress, the use of attention-based and Transformer architectures in ASR encourages the mixing of mul-

tiple levels of information within shared representations. As a result, acoustic and semantic features become tightly entangled, jointly encoding fine-grained acoustic realization, pronunciation patterns, and higher-level semantic intent. While such entanglement is often acceptable for conventional transcription, it becomes a structural limitation when distinct forms of output are required from the same speech signal. Specifically, in settings that require jointly producing surface-level transcription (speech as spoken) and meaning-oriented transcription (speech as intended), standard ASR training paradigms expose unresolved tensions (Graves & Jaitly, 2014; Chan et al., 2016). From a representation learning perspective, this arises because a single objective dominates encoder optimization, compressing heterogeneous linguistic information into a shared space where task-specific aspects cannot be selectively accessed.

To address this, multi-task learning (MTL) introduces competing supervisory signals at the shared encoder, which can encourage limited task-specific separation in the learned representations (Caruana, 1997; Ruder, 2017). However, MTL alone does not guarantee explicit or stable feature separation. Crucially, the presence of multiple, structurally distinct output objectives allows intermediate representations to be directly influenced during training, rather than analyzed only post hoc. Realizing this potential, however, requires explicit mechanisms for organizing intermediate representations.

In parallel, sparse autoencoders (SAEs) have emerged as a powerful mechanism for decomposing dense neural representations into sparse, feature-level components (Elhage et al., 2022). When integrated into a MTL framework, such structured decomposition provides a mechanism through which heterogeneous supervision signals can influence and potentially stabilize intermediate representations during training. Most existing work, however, trains SAEs post hoc for analysis purposes (Cunningham et al., 2024), leaving open whether such feature decomposition can play a functional role during model training. In this context, we use the term disentanglement to denote the functional separation of task-relevant representational components, rather than classical factorized latent disentanglement.

In this work, we propose a SAE–guided dual-output ASR

[1]Anonymous Institution, Anonymous City, Anonymous Region, Anonymous Country. Correspondence to: Anonymous Author <anon.email@domain.com>.

Preliminary work. Under review by the International Conference on Machine Learning (ICML). Do not distribute.

framework that integrates representation disentanglement directly into end-to-end training. Our model builds on a shared Conformer encoder and introduces a SAE whose latent representations are jointly constrained by alignment and generation objectives. Specifically, we apply auxiliary Connectionist Temporal Classification (CTC) losses both at the encoder output and at the SAE latent space to provide explicit alignment supervision, while employing two Transformer-based decoders for surface-level and meaning-oriented transcription. The two decoders interact through cross-attention, allowing controlled information flow between surface and semantic outputs without collapsing them into a single representation. By coupling alignment and generation constraints within a shared sparse latent space, our approach promotes functional disentanglement beyond what is typically achieved by conventional multi-task training.

We evaluate the proposed framework on dual-output speech recognition tasks that require both surface-level and meaning-oriented transcription. Our experiments compare against standard multi-task baselines that share encoder representations without explicit disentanglement. Results show that introducing alignment-aware sparse representations leads to consistent improvements across both output objectives, while yielding more structured internal representations.

Our main contributions are as follows:

- We propose an alignment-aware dual-output ASR framework that incorporates a SAE as a trainable component within end-to-end learning, enabling structured decomposition of shared speech representations beyond conventional MTL.

- We apply auxiliary CTC supervision at both the encoder output and the SAE latent space, using alignment constraints to directly shape sparse representations that support dual-output transcription.

- We show that jointly modeling surface-level and meaning-oriented transcription yields complementary outputs with strong performance on both tasks, supported by controlled interaction between dual Transformer decoders.

## 2. Related Work

### 2.1. Multi-task Learning for Speech Recognition

In end-to-end ASR, MTL has become a standard paradigm for jointly modeling alignment, acoustic, and linguistic information (Chiu et al., 2018; Prabhavalkar et al., 2023). With the emergence of high-capacity architectures such as the Conformer (Gulati et al., 2020) and the Transformer Transducer (Zhang et al., 2020), MTL approaches have been

further developed to integrate multiple training objectives within unified architectures. A widely adopted MTL strategy is to employ CTC as an auxiliary objective alongside attention-based sequence-to-sequence models. Joint CTC and attention training has been shown to improve alignment robustness and training stability, particularly for long or noisy input sequences (Kim et al., 2017; Watanabe et al., 2017). As encoder architectures have grown deeper, these approaches have been further extended into hierarchical MTL frameworks by introducing auxiliary CTC supervision at intermediate layers, leading to improved optimization stability in deep models (Nozaki & Komatsu, 2021; Djeffal et al., 2023).

### 2.2. Representation Disentanglement in Speech Models

In modern speech recognition systems, a shared acoustic encoder learns intermediate representations that encode multiple heterogeneous factors spanning acoustic, speaker, and higher-level linguistic information (Watanabe et al., 2017; Gulati et al., 2020). Prior work has analyzed how different types of information are distributed across encoder layers, showing that acoustic and linguistic information is not uniformly represented across network depth, with deeper layers exhibiting increasingly compressed and overlapping representations (Belinkov et al., 2017; Pasad et al., 2021; 2023). Such entanglement poses challenges for tasks that require selective access to distinct linguistic factors.

To address this, various disentanglement strategies have been explored in speech representation learning, including probabilistic factorization (Hsu et al., 2017), adversarial objectives (Qian et al., 2020; Wang et al., 2021a;b; Yang et al., 2022) and cross-modal self-supervision (Nagrani et al., 2020). However, these approaches often require careful architectural design or auxiliary supervision that may not transfer easily across tasks (Bengio et al., 2013; Locatello et al., 2019). More recent analyses suggest that large-scale self-supervised pretraining does not guarantee clean factor separation, as speaker and linguistic information remain intertwined across layers (van Niekerk et al., 2021; Polyak et al., 2021). These findings indicate that factor-specific control does not arise naturally from self-supervised objectives alone, but requires additional structural constraints.

Beyond probabilistic and adversarial formulations, sparsity constraints have been studied as a principled mechanism for decomposing dense and polysemantic representations. Recent theoretical work has shown that sparsity improves the identifiability of shared and task-specific features in multitask learning settings and enables the recovery of independent latent factors from complex nonlinear mixtures (Lachapelle et al., 2022; 2023). These insights have motivated the use of SAEs to analyze and decompose neural representations in settings where multiple factors of vari-

ation are jointly encoded (Olah et al., 2020; Elhage et al., 2022). While SAEs have primarily been employed as post hoc analysis tools (Cunningham et al., 2024; Templeton et al., 2024), their potential as a functional component during end-to-end training remains largely unexplored.

### 2.3. Sparse Autoencoders and Atomic Feature Decomposition

SAEs have gained renewed attention as an interpretable tool for decomposing high-dimensional representations into sparse, atomic features. They originate from early work on sparse coding, which assumed sparsity as a structural prior on latent representations (Olshausen & Field, 1996). This idea was later incorporated into neural autoencoder models by introducing sparsity constraints within trainable encoder-decoder architectures (Ranzato et al., 2007; 2008; Ng, 2011).

Bricken et al. (2023) reinterpret SAEs as a dictionary learning tool for decomposing frozen internal activations of large language models into sparse, monosemantic features, shifting their role from representation learning to post-hoc mechanistic interpretability. This perspective has motivated extensions to efficient multi-layer feature extraction (Shi et al., 2025) and prompt-based steering for in-context learning (Cho & Hockenmaier, 2025). Beyond language models, SAE-based feature decomposition has been successfully extended to vision encoders (Olson et al., 2025), multimodal representations (Lou et al., 2025; Zaigrajew et al., 2025), and diffusion models (Surkov et al., 2024). These results suggest that SAEs can extract interpretable, task-relevant features across diverse modalities. However, their application to speech representations remains unexplored. In this work, we address this gap by incorporating SAEs into a dual-output ASR framework to study their effectiveness in decomposing speech representations.

## 3. Method

We propose **SODA** (**S**parse **O**utput **D**ual-decoder **A**SR), a MTL framework that jointly performs surface-level transcription and meaning-oriented transcription within a unified end-to-end speech recognition model. Our framework addresses the task interference problem in cascaded decoders through gradient separation while leveraging sparse representations for improved feature learning. Figure 1 illustrates the overall framework.

### 3.1. Problem Formulation

Let $\mathbf{X} = (\mathbf{x}_1, \mathbf{x}_2, \ldots, \mathbf{x}_T) \in \mathbb{R}^{T \times D}$ denote an input sequence of log mel-spectrogram features, where $T$ is the number of frames, $\mathbf{x}_t \in \mathbb{R}^D$ is the acoustic feature vector at time step $t$, and $D = 80$. Our goal is to produce two output sequences: (1) a surface-level transcription $y^{\text{surf}} = (y_1^{\text{surf}}, \ldots, y_N^{\text{surf}})$ of length $N$ that represents the verbatim spoken form, and (2) a meaning-oriented transcription $y^{\text{mean}} = (y_1^{\text{mean}}, \ldots, y_M^{\text{mean}})$ of length $M$ that represents the intended written form, where each $y_i$ is a character drawn from a fixed vocabulary $\mathcal{V}$.

This dual-output formulation is motivated by the characteristics of spoken Korean, where pronunciation variations from non-native speakers or colloquial speech often diverge from standard written forms. The surface-level output preserves these spoken characteristics for verbatim transcription, while the meaning-oriented output produces normalized text suitable for downstream NLP applications such as summarization or machine translation.

### 3.2. SODA Framework

**Conformer Encoder.** We employ a Conformer encoder consisting of $L_{\text{enc}} = 12$ layers. Each Conformer block combines self-attention and convolution modules to capture both global and local dependencies in the acoustic features:

$$\mathbf{H}^{\text{enc}} = \text{ConformerEncoder}(\mathbf{X}) \in \mathbb{R}^{T' \times d} \qquad (1)$$

where $d = 256$ is the hidden dimension. The input sequence is downsampled by a factor of four through convolutional subsampling layers, resulting in $T' = T/4$.

**Sparse Autoencoder.** A key challenge in multi-task ASR is that the encoder representations must simultaneously support multiple objectives with different information requirements. Dense representations tend to encode multiple features in superposition, making it difficult for downstream heads to selectively access relevant information. To address this, we introduce a SAE that decomposes the dense encoder output into a higher-dimensional sparse representation. The SAE encoder projects $\mathbf{H}^{\text{enc}}$ into an expanded space:

$$\mathbf{Z} = \text{ReLU}(\mathbf{H}^{\text{enc}}\mathbf{W}_{\text{enc}} + \mathbf{b}_{\text{enc}}) \in \mathbb{R}^{T' \times d_{\text{sae}}} \qquad (2)$$

where $d_{\text{sae}} = 4096$ corresponds to an expansion factor of 16, allowing the model to distribute information across a higher-dimensional latent space. The ReLU activation, combined with sparsity regularization, encourages only a small subset of dimensions to be active for each input. The SAE decoder then reconstructs the original representation:

$$\hat{\mathbf{H}}^{\text{enc}} = \mathbf{Z}\mathbf{W}_{\text{dec}} + \mathbf{b}_{\text{dec}} \in \mathbb{R}^{T' \times d} \qquad (3)$$

where $\hat{\mathbf{H}}^{\text{enc}}$ is the reconstructed representation. The SAE is trained with a reconstruction loss and an L1 sparsity penalty:

$$\mathcal{L}_{\text{sae}} = \|\mathbf{H}^{\text{enc}} - \hat{\mathbf{H}}^{\text{enc}}\|_2^2 + \lambda_{\text{sparse}}\|\mathbf{Z}\|_1 \qquad (4)$$

where $\lambda_{\text{sparse}} = 0.08$. The reconstruction term ensures that $\mathbf{Z}$ preserves sufficient information, while the sparsity

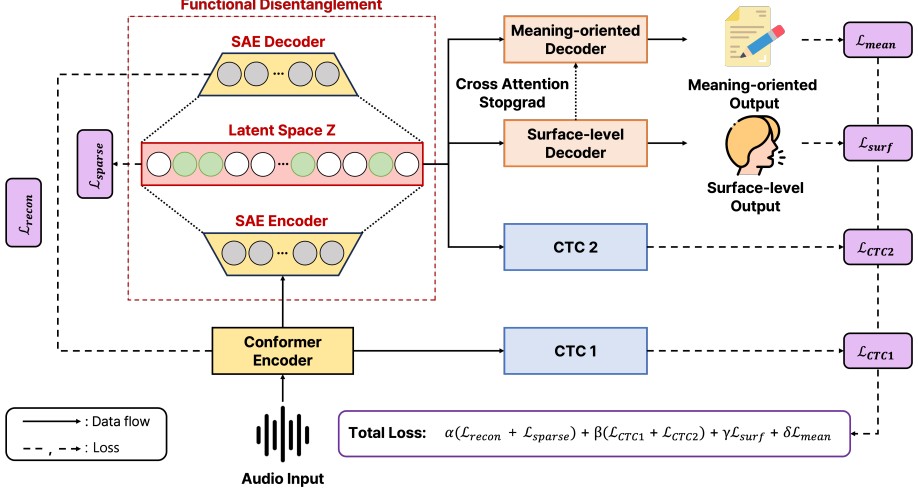

*Figure 1.* The overall framework of SODA. The Conformer encoder output is passed through a SAE to obtain disentangled representations. Dual CTC heads provide alignment supervision at both dense and sparse spaces. The Surface Decoder generates surface-level transcription, whose hidden states (with gradient separation) are passed to the Meaning Decoder for meaning-oriented transcription.

term encourages disentangled features by activating only a small subset of dimensions for each input. The sparse representation $\mathbf{Z}$ is then passed to downstream components ($\text{CTC}_2$, Surface Decoder, Meaning Decoder).

**Dual CTC Heads.** We employ two CTC (Graves et al., 2006) heads operating on different representation spaces:

$$\mathcal{L}_{\text{ctc1}} = \text{CTC}(\text{Linear}_{d \to |\mathcal{V}|}(\mathbf{H}^{\text{enc}}), y^{\text{surf}}) \qquad (5)$$

$$\mathcal{L}_{\text{ctc2}} = \text{CTC}(\text{Linear}_{d_{\text{sae}} \to |\mathcal{V}|}(\mathbf{Z}), y^{\text{surf}}) \qquad (6)$$

Both CTC heads are trained to predict the surface-level transcription $y^{\text{surf}}$, as it preserves the monotonic alignment between acoustic frames and character sequences. $\text{CTC}_1$ learns frame-level alignment in the dense acoustic space ($d = 256$), while $\text{CTC}_2$ ensures that the sparse representation ($d_{\text{sae}} = 4096$) retains sufficient information.

**Surface Decoder.** The Surface Decoder is a Transformer decoder with $L_{\text{surf}} = 8$ layers that generates the surface-level transcription by attending to the sparse representation $\mathbf{Z}$:

$$\mathbf{H}^{\text{surf}} = \text{SurfaceDecoder}(y^{\text{surf}}_{<t}, \mathbf{Z}) \in \mathbb{R}^{N \times d} \qquad (7)$$

where $N$ is the length of $y^{\text{surf}}$. We apply Pre-LayerNorm (Xiong et al., 2020) for training stability. The decoder outputs a probability distribution over the vocabulary $\mathcal{V}$:

$$P(y^{\text{surf}}|\mathbf{X}) = \prod_{t=1}^{N} \text{softmax}(\mathbf{W}_{\text{surf}}\mathbf{h}^{\text{surf}}_t) \qquad (8)$$

**Meaning Decoder with Dual Cross-Attention.** The Meaning Decoder generates the meaning-oriented transcription with access to both the sparse acoustic features and the Surface Decoder's hidden states. It consists of $L_{\text{mean}} = 4$ layers, each with a dual cross-attention mechanism that allows the model to leverage both sources of information. Let $\mathbf{h}_t$ denote the embedded representation of the token at position $t$. Each layer computes:

$$\tilde{\mathbf{h}}_t = \text{SelfAttn}(\mathbf{h}_t) \qquad (9)$$

$$\mathbf{h}^{\text{audio}}_t = \text{CrossAttn}_{\text{audio}}(\tilde{\mathbf{h}}_t, \mathbf{Z}) \qquad (10)$$

$$\mathbf{h}^{\text{text}}_t = \text{CrossAttn}_{\text{text}}(\mathbf{h}^{\text{audio}}_t, \mathbf{H}^{\text{surf}}_{\text{detach}}) \qquad (11)$$

$$\mathbf{h}'_t = \text{FFN}(\mathbf{h}^{\text{text}}_t) \qquad (12)$$

This design allows the Meaning Decoder to leverage both the original acoustic information (for disambiguation) and the intermediate transcription context (for informed generation). The subscript detach indicates gradient separation, which we discuss in Section 3.3.

### 3.3. Gradient Separation for Task Decoupling

A critical challenge in our cascaded decoder design is task interference, where gradients from the meaning-oriented loss propagate back through the Surface Decoder and corrupt the surface-level transcription objective. In preliminary experiments without gradient separation, we observed that the Surface Decoder produces outputs resembling meaning-oriented text rather than faithful surface-level transcriptions. To address this, we introduce gradient separation by detaching the hidden states before passing them to the Meaning Decoder (see Figure 1):

$$\mathbf{H}^{\text{surf}}_{\text{detach}} = \text{stopgrad}(\mathbf{H}^{\text{surf}}) \qquad (13)$$

This ensures that: (1) the Surface Decoder is optimized solely by $\mathcal{L}_{\text{surf}}$, (2) the Meaning Decoder receives transcription context but cannot influence the Surface Decoder's parameters, and (3) each decoder specializes in its designated task without interference. Importantly, the dual cross-attention still allows information flow during the forward pass—only the backward gradient flow is blocked.

### 3.4. Training Objective

The overall training objective is a weighted combination of loss terms:

$$\mathcal{L} = \alpha(\mathcal{L}_{\text{ctc1}} + \mathcal{L}_{\text{ctc2}}) + \beta\mathcal{L}_{\text{surf}} + \gamma\mathcal{L}_{\text{mean}} + \delta\mathcal{L}_{\text{sae}} \quad (14)$$

where $\mathcal{L}_{\text{surf}}$ and $\mathcal{L}_{\text{mean}}$ are cross-entropy losses with label smoothing ($\epsilon = 0.1$). The loss weights are set to $(\alpha, \beta, \gamma, \delta) = (0.15, 0.45, 0.15, 0.10)$, placing the highest emphasis on surface-level transcription as it serves as the foundation for meaning-oriented generation.

### 3.5. Inference

During inference, we employ a two-stage decoding process:

1. **Surface-level transcription**: Generate $\hat{y}^{\text{surf}}$ using beam search with the Surface Decoder.

2. **Meaning-oriented transcription**: Generate $\hat{y}^{\text{mean}}$ by conditioning on both $\mathbf{Z}$ and $\mathbf{H}^{\text{surf}}$.

The CTC outputs can optionally be used for rescoring or as an additional output for applications requiring frame-level alignment.

## 4. Experiments

### 4.1. Experimental Setup

**Dataset.** We evaluate our model on a Korean speech dataset from AI-Hub[1], specifically the "Korean Speech Data by Chinese and Japanese Native Speakers for Education" dataset. This dataset contains read speech recordings from non-native Korean speakers (L1: Chinese or Japanese) reading prompted sentences across various topics.

We use the read speech subset, where each sample contains: (1) an audio recording of the speaker reading a prompt, (2) a surface-level transcription reflecting the actual pronunciation including speech variations, and (3) the original prompt text representing the intended written form. This naturally provides paired data for our dual-output task, where surface-level transcriptions capture verbatim spoken forms

---

[1]This research used datasets from 'The Open AI Dataset Project (AI-Hub, S. Korea)'. All data information can be accessed through AI-Hub (www.aihub.or.kr).

*Table 1.* SODA hyperparameters.

| Component | Value |
|---|---|
| *Encoder* | |
| Layers / Heads | 12 / 4 |
| Hidden dim ($d$) | 256 |
| FF dim / Dropout | 1024 / 0.15 |
| *Sparse Autoencoder* | |
| Hidden dim ($d_{\text{sae}}$) | 4096 |
| Sparsity coef ($\lambda$) | 0.08 |
| *Decoders* | |
| Surface / Meaning layers | 8 / 4 |
| Dropout | 0.2 |
| *Training* | |
| Loss weights ($\alpha, \beta, \gamma, \delta$) | (0.15, 0.45, 0.15, 0.10) |
| Label smoothing | 0.1 |

and prompts serve as meaning-oriented targets. The dataset consists of 41,803 samples split into training (33,442), validation (4,180), and test (4,181) sets. The vocabulary contains 1,241 Korean characters. More information about the dataset is in Appendix A.

**Baselines.** We compare SODA against single-task and multi-task baselines. **Whisper** (Radford et al., 2022): We train in two sizes (base: 72M, small: 244M) from scratch and fine-tuned from pre-trained weights. Each model is trained separately for each task. **Conformer (single-task)**: A Conformer encoder with a single Transformer decoder, trained separately for each output type. This isolates the effect of dual-output design. **Conformer (multi-task)**: Same cascaded dual-decoder architecture as SODA, but without the SAE. This isolates the effect of SAE-based feature disentanglement.

**Implementation Details.** Table 1 summarizes the hyperparameters for SODA. The Conformer encoder consists of 12 layers with hidden dimension 256 and feed-forward dimension 1024. The SAE expands the representation to 4096 dimensions with sparsity coefficient $\lambda_{\text{sparse}} = 0.08$. The Surface Decoder has 8 layers and the Meaning Decoder has 4 layers, both using Pre-LayerNorm for training stability.

We train all models using AdamW optimizer with weight decay 0.01 and learning rate 1e-4 (1e-5 for fine-tuned models) for 50 epochs with batch size 8. We apply SpecAugment (Park et al., 2019) for data augmentation during training. Input audio is converted to 80-dimensional log mel-spectrogram features. All experiments are conducted on a single NVIDIA RTX 3090 GPU. We report Character Error Rate (CER) as the primary metric, which is standard for character-level Korean ASR evaluation.

*Table 2.* Model configurations and task performance. CER (%) reported.

| Model | Params | Surface | Meaning |
|---|---|---|---|
| *Single-task* | | | |
| Whisper-base (scratch) | 72M | 17.72 | 2.36 |
| Whisper-base (fine-tuned) | 72M | 8.75 | 48.00 |
| Whisper-small (scratch) | 244M | 20.09 | 3.51 |
| Whisper-small (fine-tuned) | 244M | 7.11 | 0.45 |
| Conformer | 35M | 12.32 | 0.75 |
| *Multi-task* | | | |
| Conformer | 37M | 11.30 | 1.05 |
| **SODA (Ours)** | 49M | 10.00 | 0.47 |

*Table 3.* Ablation study results. CER (%) reported.

| Configuration | Surface | Δ | Meaning | Δ |
|---|---|---|---|---|
| SODA | 10.00 | – | 0.47 | – |
| w/o SAE | 11.80 | +1.80 | 1.69 | +1.22 |
| w/o CTC | 22.87 | +12.87 | 0.44 | −0.03 |
| w/o Text CA | 13.67 | +3.67 | 2.20 | +1.73 |

### 4.2. Results

Table 2 presents model configurations and task performance on the test set. SODA achieves 10.00% Surface CER and 0.47% Meaning CER, demonstrating effective joint optimization of both transcription objectives in a single model.

SODA achieves competitive performance with fewer parameters. Compared to Whisper-small fine-tuned, SODA matches meaning transcription accuracy with 5× fewer parameters and no pretraining, while also improving over single-task Conformer on both tasks. Notably, most single-task baselines struggle to achieve strong performance on both tasks simultaneously. Scratch-trained models perform well on meaning transcription but poorly on surface output. Whisper-base fine-tuned shows the opposite pattern, with strong surface accuracy but catastrophic meaning performance. Only Whisper-small fine-tuned, with its large capacity and extensive pretraining, achieves balanced results. In contrast, SODA achieves comparable balance with significantly fewer parameters and no pretraining, demonstrating that joint optimization with SAE-based feature disentanglement can match the benefits of scale.

Beyond raw performance, the primary contribution of SODA lies in interpretable feature disentanglement. We analyze how the SAE learns task-specific representations in Section 5.

### 4.3. Ablation Studies

Table 3 presents ablation results analyzing the contribution of each component.

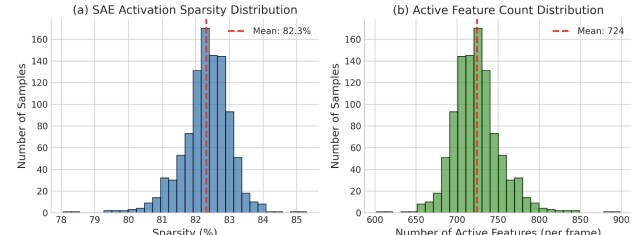

*Figure 2.* SAE sparsity analysis. (a) Sparsity distribution (mean=82.3%). (b) Active feature count per frame (mean=724).

**SAE enables effective cascaded processing.** Removing the SAE degrades performance on both tasks. Notably, this configuration performs worse than the simpler Conformer multi-task baseline, which lacks both text cross-attention and gradient separation. The cascaded architecture, where the meaning decoder attends to surface decoder outputs, requires SAE-based feature disentanglement to function effectively. Without the SAE, entangled representations propagate through the cascade, amplifying task interference rather than reducing it.

**CTC provides surface-level alignment.** Removing the CTC loss severely degrades surface transcription while leaving meaning output largely unchanged. CTC specifically targets frame-level alignment for verbatim transcription without directly contributing to meaning-oriented processing.

**Text cross-attention bridges the two decoders.** Removing text cross-attention degrades both tasks substantially. Without this connection, the meaning decoder can only access acoustic features directly, losing the benefit of linguistically processed surface representations. The surface decoder thus acts as an information bottleneck, filtering acoustic noise before meaning-oriented generation.

## 5. Representation Analysis via Feature Activations

We analyze the learned representations during test-time inference (n=4,181) to understand how SODA achieves effective dual-output transcription. Our analysis reveals that SAE-based feature separation works in conjunction with learned projection layers and the cascaded architecture to enable selective, task-appropriate processing.

**Sparsity Characteristics.** Figure 2 shows the sparsity characteristics of SAE representations. The SAE achieves a mean sparsity of 82.3%, with only 724 out of 4,096 features active per frame on average. This high sparsity enables interpretable and selective feature access for downstream tasks.

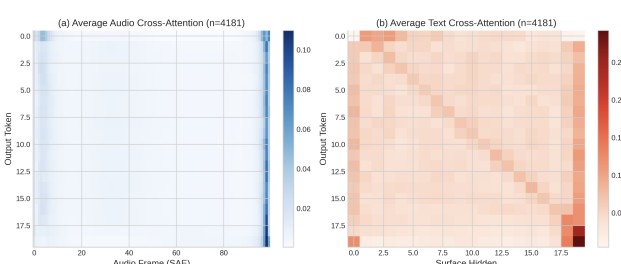

*Figure 3.* Cross-attention in the meaning decoder. (a) Audio cross-attention shows diffuse patterns. (b) Text cross-attention shows structured diagonal patterns.

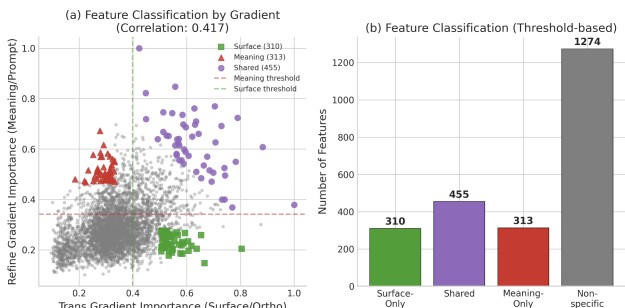

*Figure 4.* Gradient-based feature analysis. (a) Feature importance scatter plot showing surface-level (green), meaning-oriented (red), and shared (purple) features. (b) Feature classification summary.

**Cross-Attention Analysis.** Figure 3 shows cross-attention patterns in the meaning decoder. Audio cross-attention (a) exhibits diffuse, low-magnitude patterns (mean 0.007), while text cross-attention (b) shows strong, structured patterns (mean 0.050) with diagonal trends indicating monotonic refinement. Across all 4,181 test samples, text cross-attention is on average $7.8\times$ stronger than audio cross-attention. These results indicate that the meaning decoder relies primarily on surface-level representations for structured generation, while acoustic features play a secondary, supportive role.

**Task-Specific Feature Separation.** We classify SAE features using gradient-based importance measures by computing, for each feature, the gradient magnitude of the surface and meaning decoder losses. Features with high surface gradient ($>$75th percentile) but low meaning gradient ($<$50th) are classified as Surface-specific. Features with the opposite pattern are Meaning-specific, and features exceeding the 50th percentile for both are Shared. As shown in Figure 4(a), this threshold-based classification reveals clear task-specific clustering. Figure 4(b) summarizes the distribution: 310 Surface-specific, 313 Meaning-specific, 455 Shared, and 1,274 Non-specific features.

**Task-Specific Feature Selection at Projection Layers.** The gradient-based classification identifies candidate feature groups, but does not reveal *where* in the architecture

*Table 4.* Projection layer weight analysis. Mean weight magnitude by feature type. Statistical significance via Welch's t-test (Mann-Whitney U yields consistent results).

| Projection | Surface | Meaning | p-value |
|---|---|---|---|
| surface | **0.574** | 0.533 | $5.0 \times 10^{-13}$ |
| meaning | 0.497 | **0.672** | $2.1 \times 10^{-80}$ |

*Table 5.* Task specialization validation. Performance using only task-specific features (excluding Shared). Bold indicates better performance for the corresponding task.

| | Minor | | Medium | | High | |
|---|---|---|---|---|---|---|
| **Condition** | Surf. | Mean. | Surf. | Mean. | Surf. | Mean. |
| Surface only | **7.68** | 4.05 | **24.25** | 3.12 | **55.52** | 13.86 |
| Meaning only | 7.95 | **3.54** | 25.29 | **2.79** | 61.93 | **14.74** |

task-specific selection occurs. We hypothesize that the projection layers, which transform SAE representations for each decoder, learn to selectively weight different feature subsets.

Both decoders receive the same SAE latent representation Z. The Surface Decoder accesses Z through a projection layer ($4096 \to 256$), while the Meaning Decoder accesses Z through a separate projection in its audio cross-attention module. We compute the mean absolute weight magnitude for Surface-specific versus Meaning-specific features in each projection layer (Table 4).

The results reveal clear task-specific feature selection: the Surface projection preferentially weights Surface-specific features, while the Meaning projection shows even stronger preference for Meaning-specific features. Both differences are statistically significant. This demonstrates that the SAE provides a shared sparse representation, and downstream projections learn to selectively access task-relevant feature subsets without explicit supervision.

**Functional Validation of Feature Groups.** Having established *where* task-specific selection occurs, we now examine *when* these features contribute functionally. Table 6 compares performance when masking specific feature groups. To evaluate masking effects precisely, we partition test samples into two categories based on output labels: *Same* includes samples whose surface and meaning transcriptions are identical, while *Different* includes samples where the two outputs diverge. This separation isolates cases where task-specific features may play distinct roles.

For *Same* samples, masking Surface-specific or Meaning-specific features causes negligible change, while masking Shared features dramatically increases meaning error. This indicates that Shared features provide a sufficient backbone when both outputs are identical. For *Different* samples,

*Table 6.* Feature contribution by output divergence. CER (%) reported.

| Condition | Same (n=1,606) | | Different (n=2,575) | |
|---|---|---|---|---|
| | Surf. | Mean. | Surf. | Mean. |
| Baseline | 0.29 | 0.10 | 16.63 | 1.09 |
| Mask Surface | 0.24 | 0.10 | 17.47 | 1.08 |
| Mask Meaning | 0.27 | 0.10 | 16.62 | 1.17 |
| Mask Shared | 0.29 | 3.21 | 17.13 | 3.04 |

*Table 7.* Feature contribution by edit distance. Masking columns show CER change ($\Delta$) when removing the specified feature group. Isolation columns show CER using only the specified feature group (excluding Shared). Bold indicates better performance for the corresponding task.

| ED Group | n | Masking ($\Delta$CER) | | Isolation (CER) | | | |
|---|---|---|---|---|---|---|---|
| | | Surf. feat. | Mean. feat. | Surf. feat. only | | Mean. feat. only | |
| | | →Surf. | →Mean. | Surf. | Mean. | Surf. | Mean. |
| Minor (1–3) | 1,495 | +0.04 | +0.00 | **7.68** | 4.05 | 7.95 | **3.54** |
| Medium (4–10) | 875 | +1.01 | +0.25 | **24.25** | 3.12 | 25.29 | **2.79** |
| High (>10) | 205 | +3.49 | -0.14 | **55.52** | **13.86** | 61.93 | 14.74 |

masking task-specific features shows measurable but modest effects. We hypothesize that task-specific features contribute selectively depending on the degree of surface-meaning divergence.

**Scaling with Edit Distance.** The degree of divergence between surface and meaning outputs varies substantially across samples. To test whether task-specific feature contribution scales with this divergence, we stratify the *Different* samples by edit distance (Table 7). Surface-specific feature contribution scales with task difficulty: negligible for Minor, moderate for Medium, and substantial for High edit distance (ED). The isolation results reveal an interesting pattern. For Minor and Medium edit distance, the expected specialization holds: Surface-only features achieve better surface transcription, while Meaning-only features achieve better meaning transcription.

However, for High edit distance, this pattern breaks down. Surface-only features outperform Meaning-only features on *both* tasks, including meaning transcription. This confirms that when surface-meaning divergence is large, the meaning decoder cannot directly leverage Meaning-specific features effectively. Instead, the primary pathway shifts to an indirect route: Surface-specific features first improve surface output, which then propagates to the meaning decoder via text cross-attention.

**Summary.** Our analysis reveals that SODA achieves functional disentanglement through three complementary mechanisms. First, the SAE decomposes dense encoder representations into a sparse space where distinct feature groups emerge. Second, downstream projection layers learn to

selectively access task-relevant features without explicit supervision. Third, the cascaded decoder structure enables flexible information routing, with the relative importance of direct feature access versus text cross-attention shifting depending on task difficulty.

Together, these mechanisms enable adaptive dual-task processing: Shared features provide a stable backbone for low-divergence cases, while task-specific features are selectively activated as surface-meaning divergence increases.

## 6. Conclusion

We presented SODA, a dual-output ASR framework that integrates a SAE into end-to-end training for joint surface-level and meaning-oriented transcription. Unlike conventional multi-task approaches that rely solely on shared encoder representations, SODA introduces explicit mechanisms for feature separation and selective access.

Our representation analysis reveals that functional disentanglement emerges at multiple levels of the architecture. The SAE decomposes dense encoder outputs into a sparse space where distinct task-specific and shared feature groups can be identified. These features are not accessed uniformly: downstream projection layers develop selective preferences for their corresponding task-relevant features without explicit supervision. The cascaded decoder structure further enables flexible information routing, with the meaning decoder primarily leveraging surface-level representations through text cross-attention while maintaining access to acoustic features when needed. These mechanisms work together to achieve strong performance on both tasks within a single model, matching the meaning transcription accuracy of Whisper-small with significantly fewer parameters and no pretraining.

Several limitations remain and suggest directions for future work. Our evaluation is limited to a single Korean dataset with non-native speakers, and the relatively small dataset size (41K samples) may limit conclusions about scaling behavior. Extending this framework to multilingual settings would test whether the learned feature separation generalizes across languages. From a methodological perspective, the observed disentanglement emerges implicitly through multi-task training rather than being explicitly enforced. Introducing explicit disentanglement objectives, such as orthogonality constraints, may yield cleaner separation. Additionally, investigating what specific properties individual features encode could provide deeper interpretability of the learned representations.

## Broader Impact Statement

This work aims to improve speech recognition for non-native speakers, potentially benefiting language learners and cross-cultural communication. The dataset used is publicly available through AI-Hub with appropriate licensing. We do not foresee direct negative societal impacts specific to this work beyond those common to ASR technologies in general.

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

## A. Dataset Characteristics

*Table 8.* Dataset statistics with paired Surface and Meaning transcriptions.

| Split | Samples | Same (%) | Different (%) |
|---|---|---|---|
| Train | 33,442 | 38.1 | 61.9 |
| Validation | 4,180 | 38.1 | 61.9 |
| Test | 4,181 | 38.4 | 61.6 |
| **Total** | **41,803** | **38.1** | **61.9** |

*Note: Mean edit distance ratio between Surface and Meaning: 11.7%*

**Motivation for Dataset Selection.** Standard ASR benchmarks such as LibriSpeech consist of read speech from native speakers with single, normalized transcriptions. These datasets do not provide paired surface-level and meaning-oriented annotations required for our dual-output task. Our dataset contains read speech from non-native Korean speakers (L1: Chinese or Japanese), where pronunciation variations naturally occur. Each sample provides: (1) surface-level transcription reflecting actual pronunciation, and (2) the original prompt as meaning-oriented target. This pairing enables direct supervision for both outputs. Table 8 summarizes the dataset statistics: approximately 61.8% of samples exhibit differences between surface and meaning forms, with a mean edit distance ratio of 11.7%.

*Figure 5.* Examples of surface-meaning pairs showing varying divergence levels. Surface represents verbatim orthographic transcription preserving pronunciation variations, while Meaning represents normalized text with standard spelling.

| Type | Surface-level | Meaning-oriented | English Translation (Meaning) | Edit Dist |
|---|---|---|---|---|
| **Identical** | 그 식당은 음식 맛이 좋다고 한다. (Geu sikdangeun eumsik masi jot dago handa.) | 그 식당은 음식 맛이 좋다고 한다. (Geu sikdangeun eumsik masi jot dago handa.) | *They say the food at that restaurant is good.* | 0 (0%) |
| **Minor** | 나는 **핸**복하게 끝나는 영화가 **졌**다. (Naneun **haen**bokage kkeunnane un yeonghwaga **jot**da.) | 나는 **행**복하게 끝나는 영화가 **좋**다. (Naneun **haeng**bokage kkeunnan eun yeonghwaga **jo**ta.) | *I like movies that end happily.* | 2 (10.5%) |
| **Medium** | 동물을 키울 수 있는 **큰** 집에 살고 **싶어요**. (Dongmureul kiul su inneun keun j ibe salgo sipeo**yo**.) | 동물을 키울 수 있는 **크고 좋은** 집에**서** 살고 **싶어.** (Dongmureul kiul su inneun keu**g o joeun** jibe**seo** salgo sipeo.) | *I want to live in a big, nice house where I can raise animals.* | 7 (25%) |
| **High** | **싫어 그 노고 싶지 근로고 죽지** 마세요. (**Sireo geu nogo sipji geunrogo jukji ma**seyo.) | **신호등에서 길을 건넌 후 직진하**세요. (**Sinhodeungeseo gireul geonneon hu jikjinha**seyo.) | *After crossing the street at the traffic light, go straight ahead.* | 16 (80%) |

**Korean Phonological Characteristics.** Korean orthography allows near-phonetic transcription due to its phonemic writing system, making surface-meaning divergence explicitly representable. Common phenomena include liaison, tensification, and nasalization, which create systematic differences between spoken and written forms. Table 5 illustrates representative examples ranging from identical pairs to high-divergence cases where edit distance reaches 80%.

