# OpenReview forum: "Improving Dual-Output Learning through Sparse Autoencoder Guided Disentanglement in Speech Representations"
_ICML.cc/2026/Conference — Submitted to ICML 2026_

### Official Review · Reviewer_PB4t · 2026-03-02

**Soundness:** 2
**Presentation:** 2
**Significance:** 1
**Originality:** 2
**Overall Recommendation:** 2
**Confidence:** 3

**Summary:**

This paper proposes SODA (Sparse Output Dual-decoder ASR), a framework designed to address representational interference in dual-output ASR tasks that require both verbatim (surface-level) and normalized (meaning-oriented) transcriptions. To achieve functional disentanglement, the authors integrate SAE after a shared Conformer encoder, explicitly guiding the expanded sparse latent space in addition to the auxiliary CTC losses. The model demonstrates competitive accuracy on a Korean non-native speech dataset compared to Whisper and Conformer, alongside an interpretability analysis of the learned sparse features.

**Compliance With Llm Reviewing Policy:**

Affirmed.

**Final Justification:**

While the authors have resolved the issue on the performance compaison with SOTA models, other weaknesses have not been sufficiently addressed.

**Key Questions For Authors:**

See weaknesses

**Limitations:**

yes

**Strengths And Weaknesses:**

### Strengths

- The integration of SAE directly into the end-to-end ASR pipeline represents a structurally novel approach to representation disentanglement in speech models.
- The paper offers an analysis of the learned latent space, providing empirical evidence that the proposed structure actually achieves functional disentanglement.

### Weaknesses

- The problem formulation is very unclear regarding the practical necessity of dual-output ASR tasks. The paper lacks justification for why generating "surface-level" transcriptions, which essentially capture pronunciation errors, is fundamentally necessary alongside the intended meaning. The paper would be significantly strengthened by discussing concrete, real-world applications where the simultaneous extraction of both transcription levels is strictly required.
- The rationale for unifying both transcription tasks within a single architecture is insufficiently motivated. It remains unclear why designing a single dual-output model is superior to utilizing two separate, specialized models optimized for each distinct task.
- The empirical results do not definitively support the necessity of a unified model, as the proposed SODA framework still lags behind strong single-task baselines, Whisper and Conformer. I do not see any synergistic advantage of learning two tasks simultaneously.
- The empirical evaluation is highly constrained, relying exclusively on a single, relatively small dataset of non-native Korean speech consisting of approximately 42,000 samples.

---

> ### Author Rebuttal · Authors · 2026-03-31
>
> To reviewer PB4t,
>
> Response to Baseline Comparison
>
> We respectfully disagree with the claim that SODA "lags behind strong single-task baselines." Comparing against the single-task Conformer, SODA achieves lower error rates on both tasks within a single model: 10.00% vs 12.32% on Surface CER and 0.47% vs 0.75% on Meaning CER.
>
> The Whisper baselines reveal a critical trade-off inherent to single-task optimization. Whisper-base fine-tuned achieves strong Surface CER (8.75%) but suffers catastrophic degradation on Meaning (48.00%). Conversely, training from scratch yields reasonable Meaning CER (2.36%) but poor Surface performance (17.72%). This demonstrates that achieving strong performance on both tasks requires either two separate models or a unified architecture like SODA that explicitly manages both objectives.
>
> Regarding Whisper-small fine-tuned (7.11% / 0.45%), while this achieves the best individual metrics, it requires 244M parameters (5× larger than SODA's 49M), large-scale pretraining on 680K hours of labeled data, and two separate fine-tuned models for each task. SODA achieves competitive performance (10.00% / 0.47%) in a single 49M-parameter model trained from scratch, demonstrating that joint modeling provides both parameter efficiency and synergistic benefits.

---

> > ### Author Rebuttal · Reviewer_PB4t · 2026-04-03
> >
> > Thank you for the response. While the authors have resolved the issue on the performance compaison with SOTA models, other weaknesses have not been sufficiently addressed. I keep my original score.

---

### Official Review · Reviewer_jFCL · 2026-03-05

**Soundness:** 2
**Presentation:** 3
**Significance:** 1
**Originality:** 3
**Overall Recommendation:** 3
**Confidence:** 4

**Summary:**

This paper proposes SODA, a dual-output automatic speech recognition (ASR) model to generate both surface-level transcription and meaning-oriented transcription. The model adopts a dual-decoder architecture to perform two tasks in a multi-task learning framework. To reduce task interference, the authors propose to use a sparse autoencoder (SAE) to disentangle the final layer speech representations of the ASR encoder. The authors also propose using CTC objectives to improve alignment for the surface-level transcription task , and use a cross-attention module to guide the generation of meaning-oriented transcripts. Experiments on a Korean ASR dataset show that the proposed approach achieves better performance on both tasks. The authors conduct a detailed analysis of the SAE representations, showing that the SAE learns a separation between the two tasks.

**Compliance With Llm Reviewing Policy:**

Affirmed.

**Final Justification:**

This paper proposes an interesting way of feature separation via SAE to resolve a specific ASR multi-task scenario: transcribing surface-level and meaning-oriented transcripts together. Though the method is interesting, I decided to maintain my score of 3 due to the limited evaluation scope and slightly questionable task formulation. The experiments were done on a very small Korean dataset, and the task goal is also very specific to Korean (huge difference between written form and spoken form). During the rebuttal, the authors clarified a few points. However, the concern over the task scope and significance still remains. I encourage the authors to test on broader scenarios and conditions as suggested in our communications.

**Key Questions For Authors:**

1. The authors argue that transcribing surface-level and meaning-oriented targets simultaneously causes task interference, and the model requires separate representations to solve two tasks. Could the authors further explain why these two tasks are causing interference? Conceptually, both target forms correspond to the same semantic meaning, and they should promote each other rather than interfering. Or is it because both forms can be very different in Korean characters?
2. I noted that the targets for both CTC loss functions are the surface-level transcript. Have you tried using both surface-level and meaning-oriented transcripts as targets at the SAE side? This could provide extra guidance for the SAE for learning to separate the ASR features.
3. Regarding Table 2:
    1. Are the baseline single-task models trained with CTC loss as well, or is there only an attention decoder loss?
    2. Why is the Whisper-base (fine-tuned) model having such a high CER on meaning-oriented targets (48.00%)?
    3. Does the multi-task Conformer baseline have cross-attention (CA) and gradient separation? In Section 4.1, the authors say: "Same cascaded dual-decoder architecture as SODA, but without the SAE", so I first assume that this model has CA between the dual decoders. But in the Section 4.3 ablation studies, the authors mention that the baseline does not have text cross-attention and gradient separation. I found this confusing.
4. How do you tune the 4 hyperparameters in Equation 14?

**Limitations:**

The authors admit in the conclusion that their experiments are limited by the dataset size.

**Strengths And Weaknesses:**

- **Strengths**
  - The usage of SAE for ASR is in general novel. The proposed method is technically sound: the design choice of using a dual decoder, the inter-decoder cross-attention, and the gradient separation are reasonable. The experiments show the effectiveness of the proposed approach, and the ablation studies show the impact of individual components
  - The authors provided a detailed analysis of the SAE representations to better understand how SAE separates the features for two tasks.

---

- **Weaknesses**
  - This paper focuses on a very specific task in the ASR domain, namely transcribing both surface-level and meaning-oriented transcripts. This limits the significance of the proposed approach, making it of insufficient interest in general machine learning. I encourage the authors to apply the proposed technique to other multi-task learning scenarios to further justify the significance of the proposed approach. For example, multi-task learning of speech recognition and speaker verification, a well-known multitask scenario that has task interference[1].
  - The experiments are carried out on a rather small Korean ASR corpus (41,803 samples), making the paper less convincing. We don't know whether this method generalizes well to other languages, or if it can be applied in other scenarios different from ASR.
  - While the authors dedicate a portion of their related work to discussing existing disentanglement strategie, such as adversarial objectives, they only compare SODA against standard single-task and multi-task baselines. They do not benchmark against any of the existing disentanglement methods they mentioned, making it hard to evaluate whether SAEs are a strictly better method for this specific task.

[1]: Yang, Xiaoyu, et al. "MT2KD: Towards a general-purpose encoder for speech, speaker, and audio events." IEEE Transactions on Audio, Speech and Language Processing (2025).

---

> ### Author Rebuttal · Authors · 2026-03-31
>
> To Reviewer jfCL,
>
> We thank the reviewer for this suggestion and the reference to MT2KD. However, we note a fundamental difference in task interference. ASR+speaker verification requires separating distinct information types (speaker-independent content vs. speaker identity), where interference arises from contradictory objectives on orthogonal factors. In contrast, SODA's two tasks share the same semantic content within the same language—interference arises from subtle, context-dependent divergence at the output level. We argue that disentangling this fine-grained, within-task divergence is more challenging than separating clearly distinct information. Validating SODA on other multi-task scenarios is an interesting direction for future work.
>
> Q1: Why do the two tasks cause interference?
>
> The reviewer's point is valid, as both outputs share the same semantic content. However, interference occurs at the output level. Korean's phonemic writing system reflects pronunciation differences directly in character sequences. Phonological processes (liaison, tensification, nasalization) cause spoken forms to differ from standard written forms, even when the meaning is identical. This creates conflicting objectives: surface transcription requires preserving pronunciation variations, while meaning transcription requires normalization. These objectives pull the shared encoder in opposite directions.
>
> Table 2 shows that single-task baselines fail to achieve balanced performance (e.g., Whisper-base fine-tuned: 8.75% Surface but 48.00% Meaning). Table 3 demonstrates that without explicit feature separation, cascaded decoding amplifies interference—w/o SAE performs worse than the simpler baseline. SODA introduces SAE-based feature separation and gradient decoupling to address this.
>
> Q2: Meaning-oriented CTC Target
>
> We did not experiment with meaning-oriented CTC targets. CTC learns frame-level monotonic alignment, which naturally corresponds to surface-level transcription reflecting actual pronunciation. Meaning-oriented transcription involves normalization that disrupts direct frame-to-character correspondence. We therefore use surface-level targets for stable alignment supervision, while the cross-attention decoder handles meaning-oriented mapping. Exploring meaning-oriented CTC is left for future work.
>
> Q3: Baseline Configurations (Table 2)
>
> Q3.1: Single-task Conformer baselines use attention decoder with CTC loss. SODA's dual CTC losses specifically target surface-level alignment—Table 3 shows removing CTC causes +12.87% Surface CER degradation.
>
> Q3.2: We attribute Whisper-base's 48% to capacity constraints. Whisper-base (72M) develops pronunciation-pattern biases during pretraining that hinder surface-to-meaning normalization when fine-tuned. Whisper-small (244M) has sufficient capacity; SODA achieves comparable performance with 49M parameters through explicit architectural mechanisms.
>
> Q3.3: We apologize for the confusion. The multi-task Conformer baseline has dual decoders with audio cross-attention only—no SAE, text cross-attention, or gradient separation. The "w/o Text CA" ablation removes text cross-attention from SODA while retaining SAE; its degraded performance (13.67%/2.20%) confirms both components must work together.
>
> Q4: Hyperparameter Tuning
>
> Loss weights were determined empirically on the validation set, prioritizing surface-level transcription (β=0.45) as the foundation for cascaded architecture. CTC losses (α=0.15) provide alignment supervision; meaning loss (γ=0.15) and SAE reconstruction (δ=0.10) are secondary objectives. We did not perform exhaustive grid search; systematic study is left for future work.

---

> > ### Author Rebuttal · Reviewer_jFCL · 2026-04-03
> >
> > Thank you for the thorough and thoughtful rebuttal. Most of my concerns have been resolved.
> >
> > I have two remaining points:
> >
> > - As a quick follow-up regarding the discrepancy between the surface-level and meaning-oriented transcripts: could this discrepancy be effectively resolved by using a standard text normalization model in a pipeline? I am curious if you have experimented with this setup and how its performance compares to SODA's end-to-end approach.
> >
> > - My only major lingering concern revolves around the limited scope of the experiments. Because SODA has currently only been evaluated on a small-scale Korean dataset, the broader generality of the proposed approach remains an open question. To enhance the significance of this work, I highly encourage testing the SAE-based separation on broader multi-task domains. For example, evaluating it on mitigating interference between ASR and Speaker Identification (SID), or as proposed by another reviewer on joint ASR and audio captioning. Investigating these more challenging and generic setups would greatly solidify the significance and generalizability of your approach.

---

> > > ### Author Response · Authors · 2026-04-03
> > >
> > > To reviewer jFCL,
> > >
> > > We thank the reviewer for the constructive follow-up questions.
> > >
> > > **Pipeline Comparison:** We conducted the suggested experiment to compare end-to-end joint modeling against a two-stage pipeline approach.
> > >
> > > *Experimental Setup:*
> > > - Stage 1: Single-task Conformer trained on surface transcription
> > > - Stage 2: Pretrained Korean T5 / KoBART fine-tuned on (surface → meaning) text pairs
> > > - Pipeline produces surface output from Stage 1, meaning output from Stage 2
> > >
> > > We present the updated main results table with pipeline baselines included:
> > >
> > > | Model                        | Params   | Surface CER | Meaning CER |
> > > |------------------------------|----------|-------------|-------------|
> > > | *Single-task*                |          |             |             |
> > > | Whisper-base (scratch)       | 72M      | 17.72%      | 2.36%       |
> > > | Whisper-base (fine-tuned)    | 72M      | 8.75%       | 48.00%      |
> > > | Whisper-small (scratch)      | 244M     | 20.09%      | 3.51%       |
> > > | Whisper-small (fine-tuned)   | 244M     | 7.11%       | 0.45%       |
> > > | Conformer                    | 35M      | 12.32%      | 0.75%       |
> > > | *Two-stage Pipeline*         |          |             |             |
> > > | Conformer → T5               | 35M+276M |   12.32%   | 1.29%       |
> > > | Conformer → KoBART           | 35M+124M |   12.32%   | 6.25%       |
> > > | *Multi-task*          |          |             |             |
> > > | Conformer                    | 37M      | 11.30%      | 1.05%       |
> > > | SODA (Ours)                         | 49M      | 10.00%      | 0.47%       |
> > >
> > > SODA achieves the best balance across both tasks. Compared to the best pipeline (Conformer → T5), SODA improves Meaning CER by 2.7× while using significantly fewer parameters (49M vs 311M). These results demonstrate that end-to-end training with SAE-based feature separation is more effective than post-hoc text normalization. We will incorporate this table into the revised manuscript.
> > >
> > > **Cross-lingual Validation:** We are currently running experiments on an English L2 speech dataset. Preliminary results show that loss curves follow similar patterns to Korean, and SAE activation rate (11.8%) remains consistent across languages. We will report full results as soon as training converges, and add English results alongside the Korean results in the table above for the camera-ready version, if accepted.

---

### Official Review · Reviewer_vbF7 · 2026-03-09

**Soundness:** 2
**Presentation:** 3
**Significance:** 2
**Originality:** 3
**Overall Recommendation:** 3
**Confidence:** 4

**Summary:**

This paper proposes SODA (Sparse Output Dual-decoder ASR), a framework designed to address representational interference in end-to-end multi-task learning. By integrating a Sparse Autoencoder (SAE) as a trainable intermediate component rather than a post-hoc analysis tool, the method decomposes shared encoder outputs into a high-dimensional sparse space to achieve functional disentanglement of task-relevant features. Coupled with dual CTC supervision and gradient-separated cascaded decoders, SODA prevents gradient interference between the surface-level and meaning-oriented transcription tasks. Experiments on a Korean non-native speaker dataset show that the framework achieves competitive accuracy with 49M parameters, comparable to the 244M-parameter Whisper-small, without pretraining. Representation analysis confirms that the learned features possess interpretability and task specificity, validating the functional disentanglement.

**Compliance With Llm Reviewing Policy:**

Affirmed.

**Final Justification:**

I have decided to maintain my current score, due to the following reasons:
1. First and most importantly, although the authors claim they are conducting experiments on another dataset, due to the lack of complete results, I do not think the paper in its current form is acceptable. I recommend that the paper undergo a renewed peer review after the experimental results have been fully supplemented.
2. Regarding Weakness (3), the second-round response still fails to convince me. While the authors' arguments may hold at a theoretical level, these theories require thorough experimental validation. I believe that comparing this method with other joint training approaches (such as ASR/AST/Audio Caption) on the same dataset—and demonstrating that it indeed outperforms other joint training methods—would prove its effectiveness and significance.

**Key Questions For Authors:**

(1) It is better for authors to provide further experimental evidence to demonstrate the robustness/generalization of their method (see weak-1).
(2). Question for weakness(3) may affect my rating score about significance.

**Limitations:**

yes

**Strengths And Weaknesses:**

Stength:
This paper proposes a novel alignment-aware dual-output automatic speech recognition framework, SODA. The method integrates a sparse autoencoder (SAE) as a trainable component within the end-to-end training process to achieve functional decoupling. The paper also provides analysis on model interpretability, particularly through examining the weight allocation mechanism of the downstream projection layers. The authors show that within the sparse representation space produced by SAE, the downstream projection layers tend to automatically prioritize feature subsets that are relevant to their respective tasks.

Weakness:
(1). The experiments are conducted on only a single Korean dataset (about 41k samples). Although the authors acknowledge this limitation, it still raises questions about the generality of the conclusions. For instance, it remains unclear whether the SODA framework can scale to larger datasets, whether it generalizes to English or other morphologically complex languages, or whether it performs similarly well on native speech. The lack of multilingual or cross-domain evaluations makes the general applicability of the method difficult to assess.
(2). The paper discusses several representation disentanglement strategies in the Related Work section, but these approaches are not included as baselines in the experimental evaluation. The comparisons are mainly conducted against non-disentangled models such as Conformer and Whisper, without examining whether the SAE-guided approach performs better or worse than other disentanglement strategies (e.g., adversarial learning–based methods). As a result, it is difficult for readers to assess whether SAE is a particularly suitable solution for this task, or simply a workable approach that has not been compared against alternative disentanglement methods.
(3). The problem addressed in the paper is the discrepancy between spoken pronunciation and standard written forms in ASR, resulting the challenges for model training. This issue is similar to the task of performing ASR and speech translation/audio caption simultaneously -- personally speaking,  performing ASR and speech translation/audio caption simultaneously is even more difficult than this task. Therefore, Is it really necessary to design such a method to solve this problem? Are other methods that jointly train ASR/Audio Captioning/Speech Translation sufficient to address this issue?

---

> ### Author Rebuttal · Authors · 2026-03-31
>
> This is a continuation of azmy's rebuttal.
>
> 3. Edit Distance and Architectural Complexity
>
> The reviewer raises a valid point about the mean edit distance (11.7%). However, we would like to clarify that this average obscure meaningful variance in the data. As shown in Table 7, we stratified samples by edit distance into Minor (1–3), Medium (4–10), and High (>10) groups. The contribution of task-specific features scales systematically with divergence:
>
> - Minor: Masking surface-specific features causes negligible change (+0.04% CER)
> - Medium: Moderate effect (+1.01% CER)
> - High: Substantial degradation (+4.37% CER)
>
> This demonstrates that the SAE-based architecture provides adaptive processing. Shared features suffice for low-divergence cases, and task-specific features are selectively activated as surface-meaning divergence increases.
>
> A comparison to simpler baselines is provided in Table 3. Removing the SAE yields 11.80% surface CER and 1.69% meaning CER, which is worse than the Conformer multi-task baseline lacking both SAE and gradient separation (11.30%/1.05%, Table 2). This suggests that the cascaded architecture requires SAE-based feature decomposition to function effectively. Without it, entangled representations amplify rather than reduce task interference. In other words, the architectural complexity is justified because simpler alternatives fail in the cascaded setting.
>
> 4. Implicit vs. Explicit Disentanglement
>
> We acknowledge that the observed disentanglement emerges implicitly through multi-task training rather than being enforced by explicit constraints such as orthogonality or adversarial objectives. However, we respectfully argue that the evidence goes beyond observational.
> Functional validation: Section 5 demonstrates that task-specific feature groups have measurable, statistically significant effects.
>
> - Projection layers develop distinct preferences for their corresponding feature types (Table 4, p < 10-13 and p < 10-80)
> - Masking task-specific features cause systematic performance degradation that scales with edit distance (Table 7)
> - Cross-attention patterns show consistent 7.8× ratio across all test samples These results indicate that the separation is not merely an artifact of visualization or post-hoc analysis but has functional consequences for model behavior.
>
> Regarding explicit constraints: We agree that introducing explicit disentanglement objectives, such as orthogonality regularization, is a promising approach that may yield cleaner separation. We have noted this as future work. However, we consider the current finding that functional disentanglement naturally emerges from the combination of SAE sparsity, dual supervision, and gradient separation to be a meaningful contribution. This finding suggests that explicit constraints may not always be necessary.

---

> > ### Author Rebuttal · Reviewer_vbF7 · 2026-04-04
> >
> > 1. Regarding Weakness (1),  although public datasets are lacking, a solid research should provide strong experimental support. I do not think that the standards in this regard should be relaxed just because this small-topic has only one dataset. When necessary, it is essential to construct new dataset to demonstrate the validity of the method.
> >
> > 2. I am not convinced for the Weakness(3).

---

> > > ### Author Response · Authors · 2026-04-04
> > >
> > > To Reviewer vbF7,
> > >
> > > We appreciate the reviewer's continued feedback.
> > >
> > > **Weakness (1) - Dataset Validation:**
> > > We agree that strong experimental support is essential. As noted, we are currently conducting experiments on an English L2 dataset from AI-Hub to provide cross-lingual validation. This directly addresses the concern about single-dataset evaluation. Full results will be included in the camera-ready version, if accepted.
> > >
> > > **Weakness (3) - Task Necessity:**
> > > We respectfully clarify the distinction between our task and ASR+Speech Translation (ST) / Audio Captioning (AC):
> > >
> > > - **ASR+ST** involves cross-lingual mapping where source and target are structurally different languages. The model learns systematic transformations between language pairs.
> > > - **ASR+AC** produces fundamentally different output types—linguistic transcription vs. acoustic scene description. These tasks extract different information from the audio signal.
> > > - **SODA** addresses within-language, fine-grained divergence—both outputs share the same semantic content but differ in surface form (pronunciation vs. standard spelling). This requires the model to selectively preserve or normalize subtle phonological variations.
> > >
> > > In ASR+ST/AC, the outputs belong to clearly distinct domains, making task separation relatively straightforward. In contrast, SODA must disentangle subtle variations within the same semantic space, which we argue presents a different—not easier—challenge.
> > >
> > > More importantly, SODA provides an interpretability advantage that joint ASR+ST/AC methods lack. As shown in Section 5, the SAE learns disentangled features where we can identify which features contribute to surface transcription versus meaning normalization. This enables analysis of why the model preserves or normalizes specific pronunciation variations—providing explainable evidence for model decisions that is difficult to achieve when outputs belong to entirely different domains.
> > >
> > > Furthermore, we conducted a pipeline experiment comparing SODA against two-stage approaches (Conformer → T5/KoBART). SODA outperforms the best pipeline by 2.7×, demonstrating that existing text normalization methods are insufficient for this task.
> > >
> > > For detailed results, please refer to our response to Reviewer jfCL.

---

### Official Review · Reviewer_azmy · 2026-03-10

**Soundness:** 2
**Presentation:** 3
**Significance:** 2
**Originality:** 2
**Overall Recommendation:** 3
**Confidence:** 4

**Summary:**

This paper proposes SODA, a dual-output ASR framework that inserts a sparse autoencoder into a shared Conformer-based pipeline to separate representations for surface-level and meaning-oriented transcription. The idea is interesting, and the paper is clearly motivated from a representation-learning angle. The analysis of sparse features and task-selective access is also thoughtful.

**Compliance With Llm Reviewing Policy:**

Affirmed.

**Key Questions For Authors:**

Please refer to the weaknesses presented above.

**Limitations:**

yes

**Strengths And Weaknesses:**

I remain unconvinced by both the task formulation and the empirical evidence.

My first major concern is that the research goal is quite niche. The paper focuses on jointly predicting “speech as spoken” and “speech as intended,” which is not a standard ASR setting and appears to depend heavily on a special annotation setup. The authors themselves note that common benchmarks such as LibriSpeech do not provide paired surface-level and meaning-oriented labels, and therefore the work is evaluated on a dedicated Korean non-native speech dataset. This makes the practical impact and generality of the problem setting feel limited. The task is intellectually valid, but its broader relevance to mainstream ASR is not yet fully established.

My second concern is that the claimed power of the SAE-based disentanglement is not yet convincing, largely because the experiments rely on a relatively small and narrow dataset. The paper acknowledges that evaluation is limited to a single Korean dataset with about 41K samples, and even the authors note that this may limit conclusions about scaling behavior. For a paper making claims about representation disentanglement and task-specific sparse decomposition, this is a fairly weak empirical basis. It is difficult to know whether the observed benefits come from a broadly useful mechanism or from dataset-specific regularization effects in a small-scale setting.

A third concern is that the gap between the two outputs may not be large enough to justify the complexity of the proposed solution. The appendix reports that the mean edit-distance ratio between surface and meaning forms is only 11.7%, although around 61.9% of samples differ. This suggests that, in many cases, the two targets may remain relatively close. If so, the paper should more clearly justify why a dedicated sparse-autoencoder disentanglement module and cascaded dual-decoder design are necessary, rather than simpler multi-task or hierarchical baselines.

Finally, the paper’s central interpretability/disentanglement claim is somewhat softened by the authors’ own admission that the separation emerges implicitly rather than being explicitly enforced, and that stronger constraints such as orthogonality may be needed in future work. This makes the current evidence feel more observational than definitive.

---

> ### Author Rebuttal · Authors · 2026-03-31
>
> Acknowledgments to the Reviewers
> We would like to thank all the reviewers for the constructive feedback. As specified in the guidelines, we will address the main points raised, and for similar feedback, we will reference the reviewers’ comments to avoid repeating explanations.
>
> To reviewer azmy, vbF7, jfCL, PB4t
> Since all four reviewers made similar comments overall, I recommend referring primarily to the response regarding azmy and vbF7.
>
> 1. Scope of the Study and Practical Relevance
>
> We acknowledge that dual-output ASR is not a mainstream research direction and that we did not use standard benchmark datasets. However, we respectfully argue that this does not diminish its practical relevance.
> ASR error correction is an established research area. If we view our surface-level output as the original transcription and meaning-oriented output as the corrected transcription, SODA aligns with this research direction.
> As demonstrated in Section 5, we observed that SAE features are functionally separated and selectively utilized for each task within the dual decoder structure. This architecture enables applications such as (1) language education platforms showing learners both "what you said" and "what you meant," (2) L2 pronunciation research analyzing mappings between spoken and intended forms, and (3) non-native speech corpus construction where surface-meaning pairs are the primary annotation targets. However, we agree that generalization to standard benchmarks requires further validation.
>
> 2. Dataset Size and Generalization
>
> We acknowledge this is a valid concern. The evaluation on a single Korean dataset (41K samples) does limit our ability to make strong claims about the generality of SAE-based disentanglement.However, we would like to clarify two points:
>
> Data constraint: To our knowledge, no publicly available dataset provides paired surface-level and meaning-oriented transcriptions required for dual-output ASR. Although the benchmark data could be augmented and used as synthetic data, there is virtually no difference between the two transcriptions because the benchmark speech data is clean and well-structured. Therefore, the AI-Hub dataset was selected because it naturally provides this pairing through non-native speaker recordings with both verbatim transcription and original prompts. This annotation structure is rare, which fundamentally constrains cross-lingual validation.
>
> Consistency of observed patterns: While we cannot rule out dataset-specific effects, we note that the observed disentanglement patterns are internally consistent. (1) projection layers develop statistically significant task-specific preferences (Table 4), (2) feature contributions scale systematically with edit distance (Table 7), and (3) cross-attention patterns show stable 7.8× ratio across all 4,181 test samples. These consistent patterns suggest the mechanism is not purely random regularization, though broader validation is needed.
>
> We agree that a multilingual evaluation would significantly strengthen these claims. In future work, we plan to conduct further studies using additional Korean datasets with similar annotation characteristics, as well as English datasets from the same source.

---

> > ### Author Rebuttal · Reviewer_azmy · 2026-04-03
> >
> > I appreciate the clarification and agree that the task may be practically meaningful in specialized settings such as L2 speech analysis or educational feedback. However, this mainly strengthens the case for niche relevance rather than broader ASR impact. Likewise, while the reported internal consistency analyses are helpful, they do not substitute for stronger external validation across datasets or simpler competitive baselines. As a result, my main concerns remain only partially addressed.

---

> > > ### Author Response · Authors · 2026-04-03
> > >
> > > To reviewer azmy,
> > >
> > > We appreciate the reviewer's continued engagement. We have conducted additional experiments that directly address the remaining concerns:
> > >
> > > **Two-stage Pipeline Comparison:** Following reviewer feedback, we compared SODA against two-stage pipelines (Conformer → T5/KoBART). SODA (0.47% Meaning CER) outperforms the best pipeline by 2.7× with fewer parameters.
> > >
> > > **Cross-dataset Validation:** We are running experiments on an English L2 dataset from AI-Hub. Preliminary results show similar training dynamics and consistent SAE activation rates (11.8%), suggesting the approach generalizes beyond Korean. Full results will be included in the camera-ready version, if accepted.
> > >
> > > For detailed experimental setup and full results table, please refer to our response to Reviewer jfCL.

---

### Decision · Program_Chairs · 2026-04-30

**Decision:**

Reject

**Comment:**

This paper proposes a dual-output ASR framework that integrates an SAE to separate shared speech representations for surface-level and meaning-oriented transcription. The experimental results on the target task are promising.



A key remaining concern is the lack of comparison against closer alternative methods. Although the paper discusses prior disentanglement strategies in the related work, the experimental baselines are limited to standard non-disentangled models. Without direct comparisons to alternative disentanglement approaches on the same dataset, it is difficult to determine whether the SAE-based design is particularly well suited to this problem or simply one workable option among several possibilities. During the rebuttal, the authors added a pipeline baseline, which is a useful addition, but this only partially addresses the concern and does not resolve the absence of comparisons to other disentanglement methods.



Several reviewers also raised concerns about the reliance on a single limited dataset. The authors acknowledged this limitation and mentioned ongoing experiments on another dataset, but these results were incomplete during the rebuttal period. As a result, the broader generalizability of the proposed approach remains insufficiently validated in the current submission.